# Changing Decisions: The Interaction between Framing and Decoy Effects

**DOI:** 10.3390/bs13090755

**Published:** 2023-09-12

**Authors:** Adolfo Di Crosta, Anna Marin, Rocco Palumbo, Irene Ceccato, Pasquale La Malva, Matteo Gatti, Giulia Prete, Riccardo Palumbo, Nicola Mammarella, Alberto Di Domenico

**Affiliations:** 1Department of Psychological Science, Humanities and Territory, University “G. d’Annunzio” of Chieti-Pescara, 66100 Chieti, Italypasquale.lamalva@unich.it (P.L.M.); alberto.didomenico@unich.it (A.D.D.); 2Department of Medicine and Aging Sciences, University “G. d’Annunzio” of Chieti-Pescara, 66100 Chieti, Italy; 3Neuroscience Department, Boston University School of Medicine, Boston, MA 02118, USA; 4Department of Neuroscience, Imaging and Clinical Sciences, University d’Annunzio of Chieti-Pescara, 66100 Chieti, Italy

**Keywords:** cognitive bias, framing effect, decoy effect, congruent decoy, incongruent decoy, decision making, decision confidence

## Abstract

Background: Cognitive biases are popular topics in psychology and marketing, as they refer to systematic cognitive tendencies in human thinking that deviate from logical and rational reasoning. The framing effect (FE) and the decoy effect (DE) are examples of cognitive biases that can influence decision making and consumer preferences. The FE involves how options are presented, while the DE involves the addition of a third option that influences the choice between the other two options. Methods: We investigated the interaction between the FE and the DE in the case of both incongruent (ID) and congruent (CD) decoys in a sample of undergraduates (*n* = 471). The study had a two (positive vs. negative valence) × three (original, congruent decoy, incongruent decoy) within-subject design. Results: The ID option reduces the FE in both positive- and negative-framed conditions compared to the controls, while adding the CD option increases the FE only in the positive-framed condition. Additionally, the inclusion of the CD option enhances the level of decision confidence, whereas no significant differences were found in the ID condition. Conclusions: Our findings gave new insights into the interplay between two of the most frequent cognitive biases.

## 1. Introduction

Cognitive biases result from the human brain’s tendency to streamline information processing, and they refer to systematic cognitive tendencies in human thinking that deviate from logical and rational reasoning. The present study explored the interaction of two well-documented cognitive biases, namely, the framing effect and the decoy effect. Specifically, the focus was on understanding how introducing an asymmetrically dominated option (decoy) affects the framing effect and, additionally, how it impacts individuals’ confidence in their decision making. The framing effect is a common decision-making bias where a different decision is made depending on how a problem is presented [1]. Kahneman and Tversky (1979) argued that this effect occurs in a preliminary phase of the choice process where an individual identifies the value of the decision outcome as a gain or loss from a neutral reference point. Their work identified an experimental method for studying the framing effect using two pairs of decisions, presenting a risk-averse option and a risk-seeking option. Specifically, the risk-averse option refers to a low-risk alternative with lower potential benefits than a risk-seeking option, which is a high-risk alternative with higher potential benefits [2,3]. The first and most widely used task for studying the framing effect is the “Asian disease problem” [4]. In this problem, participants were presented with a hypothetical scenario where a disease is expected to kill 600 people. The participants were asked to choose the best program to fight the disease. However, a group of participants was shown two options formatted around the degree of survival from the disease (e.g., “If program A is adopted, 200 people will be saved” or “If program B is adopted, there is a 1/3 probability that 600 will be saved and a 2/3 probability that no people will be saved”). Another group of participants was shown two options formatted on the degree of mortality from the disease (e.g., “If program C is adopted, 400 people will die” or “If program D is adopted, there is a 1/3 probability that nobody will die, and a 2/3 probability that 600 people will die”). Notably, the programs presented to the two groups were equal in terms of the number of people saved (or dead) and the probability of success. Nevertheless, Tversky and Kahneman found a shift in risk preferences only due to the different descriptions (i.e., framing) of the same problem. Thus, the participants showed a tendency to be risk-averse when exposed to the survival format (positive framing), choosing to save, for sure, 200 people, and risk-seeking when exposed to the mortality format (negative framing), choosing the probabilistic option. The framing effect was also investigated in a scenario involving numerical quantities and money [4]. In this scenario, one pair of choices was framed around gaining (e.g., “a sure gain of USD 240 or a 25% chance of gaining USD 1000 and a 75% chance of gaining nothing”), while the second pair was framed around losing (e.g., “a sure loss of USD 750, or a 75% chance of losing USD 1000 and a 25% chance of losing nothing”). The results demonstrated a tendency towards risk-aversion in the gain framing condition and risk-seeking behavior in the loss framing condition. As an increasing amount of work was undertaken to better understand the ways in which the framing effect can influence decision making [5], Levin and colleagues (1998) distinguished three different types of framing effects: the risky choice framing, the attribute framing, and the goal framing [6]. Kahneman and Tversky’s classical framing experiment employs the risky choice framing effect that tests risk attitudes when a condition is framed positively or negatively. Attribute framing describes how evaluations of objects or people are perceived differently depending on whether the key attributes are described with positive or negative terminology. Finally, goal framing focuses on presenting messages that stress the positive consequences of performing an act or the negative consequences of not performing the act. In the present study, we have focused on the risky choice framing effect to test novel ways to measure and modulate its occurrence. 

The decoy effect is a phenomenon where individuals tend to have a specific change in their preference between two options when presented with a third option (the decoy) that is asymmetrically dominated. The decoy option is inferior in all respects to one of the options and only partially dominated by the other [7]. When the decoy option is present, many individuals tend to choose the dominating alternative compared to when the decoy is absent [8]. Over the years, an extensive body of research has developed around the decoy effect, given its applicability to both large-scale economic and social contexts, as well as the singular effect it has on individuals’ everyday lives. Indeed, this cognitive bias is studied in various fields such as economics and marketing [9,10,11], psychology and cognitive sciences [12], and sociology [13]. The decoy effect has been found to influence a variety of decisions from buying choices [9] to political preferences [14], hiring choices [15], health decisions [16], and choosing a romantic partner [17]. Individuals of all ages are susceptible to the decoy cognitive bias [18]. Extensive work has studied the factors influencing the decoy effect, such as the role of gender differences [19] and the role of decisiveness [20]. Recent studies provide models of rationalization with incomplete preferences that can provide further explanations for the decoy effect [21,22]. Other authors also aimed at understanding the biological underpinnings of the decoy effect by examining genetic risk factors [23] and neural biomarkers [24], with the overall goal of understanding how this cognitive bias influences decision-making mechanisms. 

Changes in decision confidence have been investigated in the context of decision-making biases such as framing and decoy effects as well as anchoring, truthfulness, and familiarity biases [25,26,27,28,29,30]. Importantly, research investigating decision confidence highlights that a low degree of confidence in a choice can result in the decision maker questioning or revisiting their decision [31]. Regarding the framing effect, Fishara and colleagues (1993) found that the way information is framed effectively influences the degree to which individuals are likely to make investment decisions. However, this study showed that decision confidence was not significantly affected by how the information was framed. Indeed, no significant difference in the subjects’ confidence ratings was detected between the negative framing and the positive framing conditions [29]. Nevertheless, Sieck and colleagues [32] found increased decision confidence when individuals were asked to write down the reasoning behind their decisions. Specifically, this effect emerged both in positive and negative framing conditions. The authors stated that the effect of the exposition of writing their reasoning boosted subjects’ beliefs that their choice was the best [32]. Additionally, Druckman and colleagues [33] have found that emotions also influence the degree of confidence when making a choice. Higher levels of distress cause individuals to be less confident both in the positive- and negative-framed scenarios, while anger causes an increase in confidence in both conditions [33]. Furthermore, previous research has also focused on the decoy’s effect on individuals’ decision confidence, and increased decision confidence has been found when the decoy is present [34]. Furthermore, Teppan and colleagues [34] have explored confidence by studying the interaction between the number of options available for making a choice and the decoy effect. They found that, while a greater set size of options decreases decision confidence, the addition of a decoy option compensates for this negative effect and helps boost confidence [35]. A recent study also showed that individuals’ decisiveness can have a moderating role in the decoy effect [21]. Another study has shown that the presence of a decoy increases the willingness to pay for consumer goods compared to the no-decoy condition [8]. 

Research has highlighted that several factors can interact with the framing effect, leading to its strengthening or weakening [36]. For example, a study found that when individuals are asked to analyze the gains and losses involved before making a choice, they become less vulnerable to the framing effect [36]. Similarly, other work has shown that when individuals are asked to describe the reasons behind their decisions, the framing effect diminishes [37]. In addition, the inclusion of contextual information seems to diminish or even eliminate the framing effect. In a study where politically credible advice was added to the scenario, individuals tended to base their choice on their beliefs rather than relying on the arbitrary information presented by the framed choice [38]. Furthermore, a recent study has found that the cognitive style used to make a decision does not influence the framing effect [39]. Yet, these findings are in discordance with previous research on framing, according to which individuals requiring a high degree of cognitive function while making a decision are less likely to be influenced by the framing of the information [40]. Seo and colleagues [41] also found that engagement in cognitive mapping before making an elaborated strategic decision overcomes the framing effect. Their work also investigated the influence of emotions on the framing effect and risk-taking. The authors found that pleasant emotions eliminate the framing effect in the realm of gains and losses, while unpleasant emotions can attenuate the framing effect only in the realm of gains [41].

To the best of our knowledge, Cheng and colleagues [42] have been the only ones who have investigated the interplay between the framing and the decoy effects. Cheng and colleagues employed the decoy effect to adapt previous empirical work conducted on framing and investigated how it interacts with other decision bias mechanisms. Their aim was to analyze how the decoy effect influences and potentially reduces the framing effect [42]. More specifically, these authors employed the traditional risky choice framing paradigm, but they added a decoy presenting three instead of two options in both the positive- and negative-framed conditions. Cheng and colleagues’ revised paradigm was the following: 

“Imagine that you face the following concurrent decisions. First, examine the three decisions; then, indicate the option you prefer. Choose between: 

(A) A sure gain of USD 240.

(B’) A 15% chance of gaining USD 1000 and an 85% chance of gaining nothing.

(B) A 25% chance of gaining USD 1000 and a 75% chance of gaining nothing.”

The addition of option B’ (aka, the decoy) in this positive-framed condition attracted the decision towards option B instead of the risk-aversive choice (option A), which would most likely be chosen if options A and B were the only options. This type of question was presented in four conditions: the positive frame condition, the negative frame condition, the positive frame condition with a decoy, and the negative frame condition with a decoy. The decoy options added in two of the conditions were created to contrast the framing effect. For this reason, both were asymmetrically dominated by the options that were least likely to be chosen in the original framing condition (the risk-aversive option in the positive frame condition and the risk-seeking option in the negative frame condition). Overall, the study showed that the addition of an asymmetrically dominated option decreased the framing effect. However, no study has looked at how the decoy can be used to enhance the framing effect. Finally, research has not yet investigated how participants’ decision confidence can be modulated in the interplay between the decoy and the framing effect. 

Our study investigated the influence of the decoy on the framing effect to better understand their mutual interaction and their impact on an individual’s confidence when making a decision. We presented participants with a series of decision-making scenarios in two opposite framing conditions (positive vs. negative). Furthermore, we manipulated the choice set, comparing three experimental conditions: (I) original, two-option, choice; (II) ID—incongruent decoy (as in Cheng and colleagues), in which the decoy and the framing pointed toward the opposite options; and (III) CD—congruent decoy, in which the decoy and the framing both point toward the same option. We hypothesized that:

H0:In the original sets, participants will prefer the risk-averse alternative in the positive frame condition, whereas they will prefer the risk-seeking alternative in the negative frame condition, supporting the classic framing effect.

Adding different decoy options, we expected to reshape the original framing effect as follows:

H1a:The effect of an incongruent decoy (ID) could decrease the choice of the expected options, both in the positive framing (risk-averse alternative) and in the negative framing (risk-seeking alternative).

H1b:The effect of the ID could decrease the confidence with which participants will choose the expected options in both frame conditions.

H2a:A congruent decoy (CD) effect could increase the choice of the expected options in both frame conditions. 

H2b:The effect of the CD could increase the confidence with which participants will choose the expected options in both frame conditions.

## 2. Materials and Methods

### 2.1. Participant

A sample of 471 participants (348 female) between 18 and 35 years old (M = 22.27, SD = 4.02) was enrolled in the present study. They were all students at the “G. d’Annunzio” University of Chieti-Pescara (Italy). They provided informed consent before starting the experiment. The participants did not receive any compensation, based on prior research conducted on the same topic [42], as well as recent evidence indicating that the provision or absence of incentives does not affect the bias in participant responses [43]. The data were collected using the Qualtrics’ survey software. The Institutional Review Board of Psychology (IRBP) approval was obtained by the “G. d’Annunzio” University institutional ethical committee.

### 2.2. Design, Procedure, and Material

As explained in the introduction, the decoy effect occurs when a third asymmetrically dominated option increases individuals’ choice for the dominating option compared to another option. Therefore, we defined a congruent decoy (CD) as a third option asymmetrically dominated by the option that individuals prefer in a classic framing task, based on the frame condition (positive or negative). Conversely, we defined an incongruent decoy (ID) as a third option asymmetrically dominated by the option that individuals tend to reject in a classic framing task. We used a two (frame: positive versus negative) × three (choice set: original, congruent decoy, incongruent decoy) within-subjects design to investigate the bi-directional influence of the decoy effect on the framing effect. We had six different experimental conditions, and each one was repeated twice in a randomized way. Therefore, the participants were presented with twelve decision problems. Also, we had six scenarios (e.g., disease problem; for details, see the Materials section below); thus, the same scenario was used for two different experimental conditions. Three stimulus lists (A, B, C) were created to ensure that each scenario was equally presented twice in the six experimental conditions. Therefore, each list contained twelve problems where two repetitions of each experimental condition were presented, varying the scenario arenas. The administration of the three different lists was balanced between experimental subjects. They were asked to choose what they believed was the best option among the choices set for each problem. After that, the participants rated their confidence in their choice answering the question “How confident are you in the choice you just made?” on a scale ranging from one (extremely not confident) to six (extremely confident). 

We selected and adapted six different decision problems previously used in past studies on the framing effect: Crew, Pregnancy, and Cab [44]; Fatal Disease [45]; Money [4]; and Home Selling problem [46]. We aimed to differentiate the scenarios to avoid a possible repetition effect of the same problem, which could have encouraged consistency across subjects’ responses [47,48,49,50]. Table 1 shows an example of a scenario (Crew problem) presented in the six experimental conditions. 

## 3. Results

Chi-squared tests were performed to test our hypotheses on the differences in participants’ choices across the six experimental conditions [51]. First, we investigated the presence of the framing effect in the original set conditions. These analyses revealed that in the positively framed original set, participants preferred the risk-averse choice (70.2%) compared to the risk-seeking choice (29.8%), χ^2^ (1) = 307.34, *p* < 0.001. On the other hand, in the negatively framed original set, participants preferred the risk-seeking choice (72.7%) compared to the risk-averse choice (27.3%), χ^2^ (1) = 388.12, *p* < 0.001. Consequently, the percentage of risk-averse choices was statistically different in the two original set conditions, χ^2^ (1) = 346.77, *p* < 0.001. In other words, participants preferred the risk-averse alternative in the positive original condition, whereas they preferred the risk-seeking alternative in the negative original condition. This result supported H0 and was consistent with the classic framing effect [4]. Second, to investigate the influence of the decoy in reshaping the original framing effect, we compared the percentage of risk-averse (and risk-seeking) choices obtained in the original set conditions with those obtained in the decoy conditions. For this purpose, as in Cheng and colleagues’ study (2012), we calculate the percentage of risk-averse choices in decoy conditions, filtering out for all responses in which the decoy option was selected. Regarding the effect of the ID option, we found that the percentage of risk-averse choices in the positive original condition (70.2%) significantly decreased in the positive ID condition (54.2%), χ^2^ (1) = 47.93, *p* < 0.001. In this condition, the percentage of risk-averse choices (54.2%) was still significantly higher than the percentage of risk-seeking choices (45.8%), χ^2^ (1) = 11.54, *p* < 0.001. In the negative ID condition, the percentage of risk-seeking choices was lower (52.3%) than in the negative original condition (72.7%), χ^2^ (1) = 81.93, *p* < 0.001, indicating that the presence of a decoy reduced the choice of the expected option given the framing condition. Additionally, in the negative ID condition, the preference for the risk-seeking choice (52.3%) over the risk-averse choice (47.7%) remained significant (even if reduced), χ^2^ (1) = 3.81, *p* = 0.05. Overall, these results supported H1, revealing that the presence of an ID decreased the frequency of choices of the expected option in both the positive and negative framing conditions. Notably, for the ID conditions, we replicated the results obtained by Cheng and colleagues (2012). Regarding the effect of the CD option, we found that the percentage of risk-averse choices in the positive original condition (70.2%) was significantly higher in the positive CD condition (79.3%), χ^2^ (1) = 20.38, *p* < 0.001. As expected, the percentage of risk-averse choices was significantly higher than the percentage of risk-seeking choices (20.7%), χ^2^ (1) = 631.51, *p* < 0.001. Interestingly, no significant differences emerged in the percentage of risk-seeking choices between the negative original condition (72.7%) and the negative CD condition (76.0%), χ^2^ (1) = 2.503, *p* = 0.11. However, when comparing the percentage of risk-averse choices (24.0%) with the percentage of risk-seeking choices (76.0%) in the negative CD condition, a significant difference emerged, χ^2^ (1) = 445.89, *p* < 0.001. These results partially supported H2, revealing that the presence of CD increased the frequency of choices of the expected option only in the positive framing. A summary of the percentages of choices obtained for the six experimental conditions is shown in Figure 1. Furthermore, to explore the potential role of gender, we repeated these analyses, separating male and female participants. The results showed that no gender-based distinctions emerged in either the original or CD conditions, whereas gender differences were trivial for the ID conditions (see Appendix A). Finally, we repeated chi-square analyses separating each of the six scenarios included in the present study, showing a similar pattern in either the original or CD conditions, whereas trivial fluctuations emerged in the ID conditions (see Appendix A).

Moreover, we analyzed the number of responses when the decoy option was selected, which were excluded from the previous analyses. It is critical to note that the decoy represents an alternative that is disadvantageous compared to the dominating option. Regardless, participants can sometimes choose the decoy option. We found that the irrational choice of the decoy occurred more often when the decoy was framed as a risk-seeking alternative compared to when it was framed as a risk-averse alternative. This happened both in the positive framing (positive CD: 2.3% vs. positive ID: 13.2%), χ^2^ (1) = 78.23, *p* < 0.001, and in the negative framing (negative-CD, 12.4% vs. negative-ID, 4.4%), χ^2^ (1) = 36.16, *p* < 0.001 (see Appendix A for a graphical representation of these results). Furthermore, we conducted two one-way ANOVAs, respectively, for positive and negative framing to compare the effect of the choice set (original, ID, CD) on participants’ confidence ratings. Regarding the positive framing, we found a significant effect of the choice set on decision confidence ratings, F (2183) = 26.75, *p* < 0.001. Post hoc comparisons using the Tukey HSD test highlighted no significant differences (*p* = 0.815) between the original condition (M = 3.96, SD = 2.96) and the ID condition (M = 4.00, SD = 2.96). However, we found a significant increase (*p* < 0.001) in the confidence ratings in the CD condition (M = 4.37, SD = 2.96) compared to the original condition. Regarding the negative framing, we also found a significant effect of the choice set on confidence ratings, F (2178) = 14.84, *p* < 0.001. Post hoc comparisons using the Tukey HSD test highlighted no significant differences (*p* = 0.479) between the original condition (M = 3.84, SD = 2.96) and the ID condition (M = 3.92, SD = 2.96). However, we found a significant increase (*p* < 0.001) in the confidence ratings in the CD condition (M = 4.17, SD = 2.96) compared to the original condition. Overall, these results did not support H1b, since no significant differences were detected in the decision confidence ratings after the expected choices were selected between the original condition and ID condition, both in the positive framing and negative framing. Conversely, H2b was supported since significant differences emerged in the decision confidence ratings after the expected choices were selected between the original condition and the CD condition, both in the positive and negative framing conditions. Indeed, the presence of a congruent decoy option increased the decision confidence with which participants chose the expected options in both frame conditions.

## 4. Discussion

The current study examined the interplay between two widely known cognitive biases: the framing effect and the decoy effect. Specifically, we examined the influence of an asymmetric dominated option (decoy) on the framing effect. We also examined how the addition of the decoy influences individuals’ confidence in their decisions. Previous research shows a decrease in the framing effect when an incongruent decoy option was added. Our aim was to expand these previous findings. At the same time, we also investigated how a congruent decoy influences the framing effect. An assessment of the bi-directional influence of the decoy on the framing effect can help explain the impact of their interaction in real life. as well as help understand how other decision-making biases occur. Overall, the present findings showed that the incongruent decoy option reduced the framing effect both in the positive frame and negative frame conditions. Furthermore, the congruent decoy increased the framing effect, but only in the positive frame condition. Additionally, participants showed an increase in their choice confidence both in the positive and in the negative frame congruent decoy conditions compared to the original frame conditions. Conversely, they did not show any change in their decision confidence for the incongruent decoy conditions compared to the original ones. The present findings were consistent with previous studies on the framing effect. Indeed, we replicated the results of the original framing effect in a within-subjects design [47,52]. We replicated the results obtained by Cheng and colleagues by investigating the condition of incongruency between the decoy and the framing effect. Specifically, we found a reduction in the framing effect by 16% in the positive frame condition and by 20.4% in the negative frame condition when an incongruent decoy was added. Notably, we presented the same experimental condition twice to each subject, but without showing the same scenario problem to the participant, to avoid a possible effect of repetitions [47]. 

The present findings bring a novel overview of how the congruent decoy influences the framing effect, which is something that, to the best of our knowledge, was never investigated before. On the one side, by adding the CD option, we found an increase of 9.1% in the framing effect in the positive frame condition. On the other side, no significant difference was found in the negative frame condition when the CD option was added. We do not have a univocal explanation for this finding. Future studies should replicate and extend this difference between the positively and negatively framed scenarios when a CD option is added. However, we noted that the negative CD condition was one of the conditions with a higher number (12.4%) of decoy responses (i.e., the number of responses when the decoy option was selected). Specifically, we found that the irrational choice of selecting a decoy response occurred more often when the decoy was framed as a risk-seeking alternative compared to when it was framed as a risk-averse alternative. Therefore, we can hypothesize that adding the CD in the negative frame condition (i.e., a further risk-seeking alternative) might not make the choice easier for respondents. This possible view is in line with research pointing out that the modality of presentation of the decoy can have a substantial role in influencing the choice [53,54]. As previously discussed, both the decoy and the framing effect have been separately investigated to assess how psychological and social contextual factors can affect their decision bias. However, no framework has yet been established to understand their interaction and potentially assess how different contextual factors can influence their relationship. For this reason, future work should aim at understanding the underlying processes resulting in the decoy congruency and incongruency effect on the framing. Additionally, future investigations should compare the role of the decoy in the risky choice framing with other types of framing mechanisms such as attribute framing and goal framing [6]. 

Our study expanded the investigation of the interplay between the decoy and the framing effect by considering decision confidence. Indeed, we assessed the change in participants’ confidence in their choices when a decoy option was added. We did not observe a change in confidence with the addition of the ID option compared to the original sets. Conversely, we found an increase in choice decision confidence with the CD options compared to the original framing conditions. This phenomenon suggests that individuals’ decision confidence in the choice made increases when the decoy option is present and favors the framed option. Many studies have looked at the level of confidence during decision-making processes [29]. However, none of these studies focused on decision confidence after choices in the context of the interplay between decoy and framing effects. Our study represents the first attempt to investigate changes in decision confidence during different framed choices that occur with the manipulation of the decoy options. In this context, our findings highlight the importance of investigating people’s confidence in their decisions. Adding a decoy in favor of the framing option increases confidence and could further enhance the framing effect, leading to a longer duration and consistency over time. Accordingly, previous research investigating decision confidence highlighted that a low level of confidence in a decision could lead the decision maker to question or revisit that decision [31]. We acknowledge that a large portion of participants in our study were female and there is a large body of literature that confirms the interaction between gender and cognitive bias [55]. To partially overcome this limitation, we conducted a specific analysis on gender, showing no gender differences in either the original or CD conditions; however, trivial differences emerged in the ID conditions. Notably, in ID conditions, the role of the decoy was to mitigate the framing effect, and this trend was observed for both male and female participants. However, future studies properly targeting gender differences are needed. Likewise, we found trivial differences only for the ID conditions comparing the percentage of choices between each of the six scenarios. We do not have a univocal explanation for these differences; however, the means are mostly in the expected direction, indicating that the framing effect is at least partially reduced by the addition of an incongruent decoy option. While we acknowledge that different scenarios may elicit partly different styles of choices, our experimental paradigm was not created to test this hypothesis, and different scenarios were mainly adopted to reduce repetition bias and boredom. Future studies could delve into this issue by directly manipulating the scenario used—for instance, comparing social and monetary decisions [56]. Additionally, compared to previous studies [7], a relatively large portion of participants unexpectedly chose the decoy option, which is by definition totally unfavorable compared to the other options. However, we noted that this only happens in the two conditions where the decoy was expressed in a probabilistic form (i.e., positive ID and negative CD). This could be because participants sometimes got confused between the two probabilistic options (i.e., the decoy option and the dominant option), choosing the unfavorable one. On the one hand, this phenomenon provides further support for previous studies that have shown that even subtle changes in the form in which the decoy is presented can influence participants’ choices [57]; however, it also raises the question of whether participants paid sufficient attention in examining the probabilistic options. Future studies could delve into this issue—for example, by introducing attention checks into the experimental paradigm. Beyond these caveats, the present study contributes to exploring how the framing and the decoy effect interact, carrying implications that span from the strategic marketing efforts of businesses to the understanding of human behavior and decision making in broader economic and social contexts. For instance, our findings could have an important implication in influencing important real-life decision-making problems [58,59,60]. The addition of a third asymmetrically dominated option favoring the framed option will influence the degree of confidence in the decision taken. The decoy in favor of the framing effect option might lead to greater certainty in decision making. This phenomenon has strong potential implications for the perceived effectiveness of a decision, maybe even when very important decisions might be involved (e.g., saving human lives). From a business standpoint, understanding how framing and decoy effects interact can provide valuable insights into consumer behavior. Companies can strategically utilize these insights to design marketing campaigns and product presentations that guide consumers toward choices. By leveraging the synergy between these two effects, businesses can potentially enhance the perceived value of their offerings and influence purchasing decisions. These considerations remark on the importance of new research looking at the mutual influence of the decoy with the framing effect. 

## 5. Conclusions

The present work investigated the bi-directional role of the decoy in the framing effect. By implementing a within-subjects design, we found that the ID option reduces the framing effect both in the positive frame and negative frame conditions. However, we not only considered the decrease of the framing effect but also investigated how it can be increased. We found that adding a CD option increases the framing effect only in the positive frame condition. Furthermore, we found that CD options increase decision confidence in both positive and negative framing conditions, whereas no differences were found when ID options were added. Finally, this study has shown that the within-subjects design we created by involving different scenarios effectively studies the interaction between the decoy and the framing effect. Importantly, the within-subjects design can lead to increased statistical power since participants contribute data across multiple conditions. Moreover, it can enhance the detection of subtle changes or effects within individuals, making it well-suited for studies focused on individual differences. Therefore, future research could use a similar design to directly assess which individual characteristics could influence the tendency to change decisions due to the addition of the decoy. For example, characteristics such as the type of personality, the degree of impulsivity, and logical and cognitive abilities could play an important role. Additionally, individual differences in confidence may moderate the effectiveness of the framing/decoy effects and the susceptibility to cognitive biases such that people who are more self-confident may be more (or less) prone to be biased by the contextual information, such as the framing of the problem and the presence of decoy options. Finally, this study can be one of the first to further expand our knowledge of how different decision-making biases that affect human choice behavior can interact. Future studies can deepen this topic by considering other different decision-making biases to investigate their mutual influence on each other. 

## Figures and Tables

**Figure 1 behavsci-13-00755-f001:**
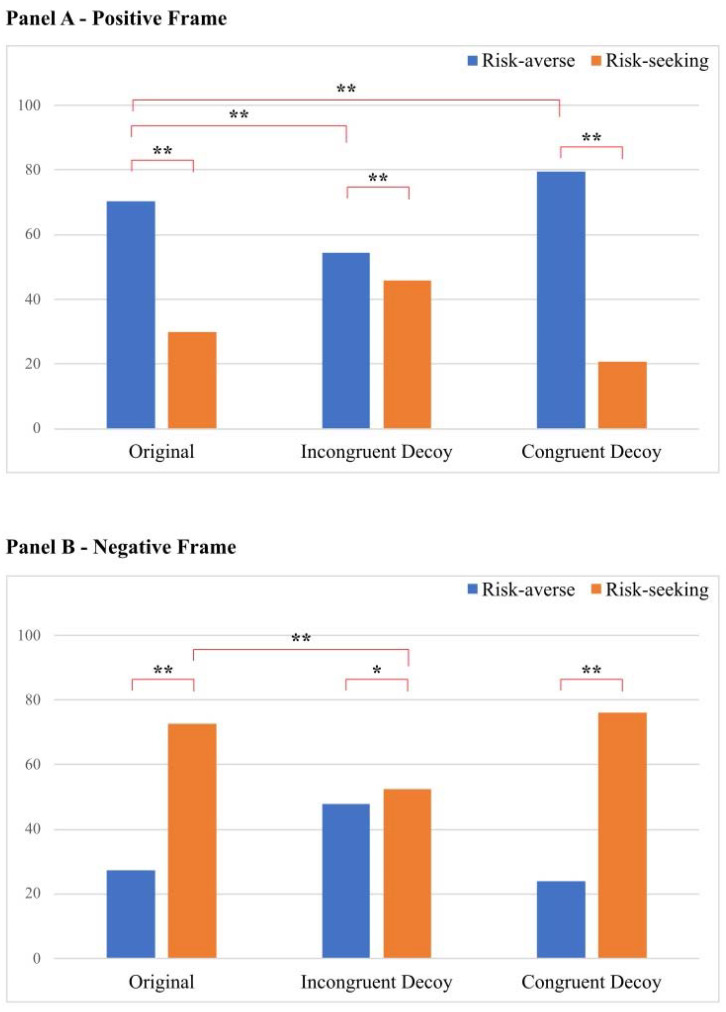
Frequency of choice in the three choice sets, separately for the positive frame (**Panel A**) and the negative frame (**Panel B**) conditions. Note. * *p*< 0.05, ** *p* < 0.001. The frequency of choice of the decoy option is not represented. Only the significant comparisons that have been discussed in the Results section are represented.

**Table 1 behavsci-13-00755-t001:** Example of an alternative of choices presented in the experimental conditions (choice set × frame) for the scenario “Crew”.

**Storyline**	**A ship hit a water mine in the middle of the ocean. There are 1200 crewmen on the ship. Their lives are in danger.** **Two (“three” in the decoy conditions) options are proposed. Assume that the estimates of the consequences are the following, and indicate the option you prefer.** **Choose between:**	
	Positive	Negative
Original	A. 400 crewmen will be saved for sure. *	A. 800 crewmen will die for sure.
	B. There is a 1/3 chance that 1200 crewmen will be saved and a 2/3 chance that nobody will be saved.	B. There is a 1/3 chance that nobody will die and a 2/3 chance that 1200 crewmen will die. *
CD	A. 400 crewmen will be saved for sure.	A. 800 crewmen will die for sure.
	A’. 300 crewmen will be saved for sure	B’. There is a 1/3 chance that 300 crewmen will die and a 2/3 chance that 1200 crewmen will die.
	B. There is a 1/3 chance that 1200 crewmen will be saved and a 2/3 chance that nobody will be saved.	B. There is a 1/3 chance that nobody will die and a 2/3 chance that 1200 crewmen will die.
ID	A. 400 crewmen will be saved for sure.	A. 800 crewmen will die for sure.
	B’. There is a 1/3 chance that 900 crewmen will be saved and a 2/3 chance that nobody will be saved	A’. 900 crewmen will die for sure.
	B. There is a 1/3 chance that 1200 crewmen will be saved and a 2/3 chance that nobody will be saved.	B. There is a 1/3 chance that nobody will die and a 2/3 chance that 1200 crewmen will die.

**Note**. Alternatives of choice were presented in randomized order. * Expected alternative of choice based on the original framing effect.

## Data Availability

The data presented in this study are available on request from the corresponding author.

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
