# Peer review of "Changing Decisions: The Interaction between Framing and Decoy Effects"

_behavsci, 2023, doi:10.3390/bs13090755_

Round 1
Reviewer 1 Report
Review for: Changing decisions: the interaction between Framing and De- 2 coy effects
Evaluation- The research question is moderately interesting and deserves exploration. The method is generally well-conducted, and the portions of the results that aren't novel replicate existing findings. New results are interesting. Overall, assuming all my comments are addressed, this paper could be a great fit for “Behavioral Sciences.”
Major comments-
1. A relatively large portion of participants chose the decoy effect (e.g., 13.2% in the positive-ID condition compared to between 3%-5% in other, similar studies such as Huber et al. 1982). This raises the question of whether participants paid sufficient attention to the questions. I'm aware that non-incentivized experiments are common in this area, but did you include any attention checks?
2. A large portion of the participants were women; this made me concerned about selection bias. Please at least control for gender in your analysis.
3. You did not display data for each vignette and the experiment was not pre-registered. Coupled with the surprising result that no significant difference was found in the negative-frame condition when the CD option was added, this makes me concerned about the replicability of the latter result.
4. I believe you should also examine the classic interaction between the two effects; specifically, comparing the size of the ID in positive framing to the CD in negative framing (and perhaps vice versa).
5. Most importantly, the effect of the confidant on the two main effects (i.e., as a potential moderator) is much more intriguing, in my opinion, than the impact of the attraction/framing on confidence that you investigated.
6. I misses the implications of your studies, for example, to business, economists or individuals.
One way to address most of my comments simultaneously is to replicate the experiment with a more balanced sample, use incentivized questions (or at least include attention checks), and pre-register.
Minor comments-
1. Line 1-71, you should describe the ideas there from the general to the most relevant and not as you did.
2. In the overview of the decoy effect please mention its axiomatic rationale (Barokas, 2017; Natenzon, 2019).
3. The use of references is not always accurate (e.g., 52-57).
4. Reference #1 us unnecessary bold.
5. I did not understand what the red lines in the figures mean, I would prefer to see confidence intervals.
6. Lines 98,346, and 367 have unnecessary space.
7. Please consider your word choice in line 278.
8. “However” should be “nevertheless” in line 99
9. Sieck misses a reference in line 100.
1. There are many unnecessary repetitions, please try not to repeat yourself.
1 Good luck in your publication journey.
Reviewer 2 Report
This paper present interesting results on two examples of cognitive bias using an experiment. It tries to assess the interaction between the decoy effect and the framing effect. Authors used a 2x3 within-subjects design and they find a that the incongruent decoy setting reduces the framing effect in both negative and positive conditions, while the congruent decoy setting increase the framing effect only in the positive condition.
The manuscript gives interesting insight on the related topic. I believe that it well assessed the related literature and the methodology. Moreover, the discussion of the results it's in line with the findings and well described.
Some comments below, specifically referred to the results and methods section.
- The sample of the study is mainly composed by female. There is a large body of literature that confirm the interaction between gender and cognitive bias. Some papers also look at the gender bias applied to the decoy effect. In the manuscript, analysis that consider the individual characteristics of the participants are missed. Also if the design is within-subject, eliminating the gender effect between the treatments, the general result can be pushed by the fact that around the 75% of the participants are female. A robustness check on this might eliminate the doubt.
- I believe that the within-subjects design is used in the right way. However, given that authors state that the design chosen was effective, I believe that more motivation about why using the within design might be more effective with respect to the between design.
- Results: also if the results are well described and the statistics are appropriated, I believe that a graphic representation of the percentage of decoy choices can improve the understanding of the results.
- It can be interesting to see what happened using different scenarios. Are the results robust separating decision by scenarios? Moreover, given the fact that participants are not compensated (useful to explain why this choice, and why neither a participation fee was given), the money scenario might have different implication if real money were involved. Discuss and motivate it.
Minor comments:
- The example from Cheng and colleagues, at the end of page 3, they (or you) used B' as the decoy alternative. However, at page 4 row 193, you use A' to describe the decoy alternative. I understood the difference only once I had a look at table 1. I think, to no create misunderstandings in the readers, that it will be better to homogenizing the letters of the decoy alternative before the table.
- There are minor typos along the paper, please, revise them (e.g. row 309, there is a point instead of a comma in the middle of the sentence).
I think the English if fine and compressible. Some minor typos to be corrected.
Round 2
Reviewer 1 Report
The authors have addressed the vast majority of my comments in a satisfactory manner. There are a few minor issues remaining:
1. I believe that the statement, "We hypothesize that the increased selection of the decoy in our study compared to previous studies may be influenced by the fact that the decoy option was frequently presented within a series of probabilistic choices, potentially complicating the decision-making process for respondents," does not serve as a satisfactory explanation. Choosing more difficult tasks that lead to the selection of a decoy introduces noise, rather than alleviating concerns regarding the replicability of the results. In contrast, the amount of time taken to complete the experiment could have a significant impact. Have you attempted to control for this factor?
2. Regarding my fourth point (and I apologize for any confusion), I intended to suggest examining whether the CD (ID) had a stronger effect in negative or in the positive framing.
3. I recommend linking your study to the recently published work by Barokas and Gamliel (2023), who also investigated the attraction effect under CD and ID conditions and assessed confidence levels (although they did not analyse the framing effect).
4. In the new version, there is a typo in reference 40.
Reference:
Barokas G, and Gamliel E. "The Moderating Role of Decisiveness in the Attraction Effect." The B.E. Journal of Economic Analysis & Policy.
